# Green Synthesis of Au Magnetic Nanocomposites Using Waste Chestnut Skins and Their Application as a Peroxidase Mimic Nanozyme Electrochemical Sensing Platform for Sodium Nitrite

**DOI:** 10.3390/foods12193665

**Published:** 2023-10-05

**Authors:** Huanan Guan, Ke Xing, Shuping Liu

**Affiliations:** 1School of Gain Science and Technology, Jiangsu University of Science and Technology, Zhenjiang 212000, China; 2College of Food Engineering, Harbin University of Commerce, Harbin 150076, China; hebcuedu@163.com; 3College of Tourism and Culinary Science, Harbin University of Commerce, Harbin 150028, China

**Keywords:** Au magnetic nanocomposites (Au@Fe_3_O_4_), electrochemical detection, green synthesis, peroxidase mimic, sodium nitrite

## Abstract

An electrochemical sensor with high sensitivity for the detection of sodium nitrite was constructed based on the peroxidase-like activity of Au magnetic nanocomposites (Au@Fe_3_O_4_). The Au@Fe_3_O_4_ composite nanoparticles were green-synthesized via the reduction of gold nanoparticles (AuNPs) from waste chestnut skins combined with the sonochemical method. The nanoparticles have both the recoverability of Fe_3_O_4_ and the advantage of being able to amplify electrical signals. Furthermore, the synergistic effect of green reduction and sonochemical synthesis provides a functional approach for the preparation of Au@Fe_3_O_4_ with significant peroxidase-like activities. The physicochemical properties were characterized using transmission electron microscopy (TEM), energy dispersive X-ray spectroscopy (EDS), the Brunauer–Emmett–Teller (BET) method, and Fourier transform infrared spectroscopy (FT-IR). The electrochemical properties of sodium nitrite were determined with cyclic voltammetry (CV) and chronoamperometry (*i-t*). The results revealed that Au@Fe_3_O_4_ acted as a peroxidase mimic to decompose hydrogen peroxide to produce free radicals, while ·OH was the primary free radical that promoted the oxidation of sodium nitrite. With the optimal detection system, the constructed electrochemical sensor had a high sensitivity for sodium nitrite detection. In addition, the current response had a good linear relationship with the sodium nitrite concentration in the range of 0.01–100 mmol/L. The regression equation of the working curve was *y* = 1.0752*x* + 4.4728 (*R*^2^ = 0.9949), and the LOD was 0.867 μmol/L (*S/N* = 3). Meanwhile, the constructed detection system was outstanding in terms of recovery and anti-interference and had a good detection stability of more than 96.59%. The sensor has been successfully applied to a variety of real samples. In view of this, the proposed novel electrochemical analysis method has great prospects for application in the fields of food quality and environmental testing.

## 1. Introduction

Nitrite is an intermediate oxidation state between nitrate and ammonia [1]. It is well known that sodium nitrite is widely used as a preservative and color fixative in additives [2,3]. Previous research has demonstrated that sodium nitrite inhibits the growth of harmful bacteria, specifically clostridium botulinum spores [4]. However, sodium nitrite irreversibly binds to hemoglobin, producing methemoglobin and further reducing the blood’s oxygen-carrying capacity [5]. The potential formation of carcinogenic N-nitrosamines is another essential aspect of food safety associated with the consumption of sodium nitrite [6]. Notably, sodium nitrite is not only found in food additives but also in animal manure and decaying organic matter, especially in soils that have been over-fertilized [7]. The World Health Organization (WHO) has set a limit of 3 mg/L for nitrite ions in potable water due to their potential toxicity [8]. Consequently, the presence of excessive sodium nitrite in food and the environment poses a grave danger to human health. To protect the natural environment and public health, the detection of excess sodium nitrite in food must be efficient, real-time, and precise. In the majority of investigations on sodium nitrite detection, fluorometry, spectrophotometry, and liquid chromatography [9,10,11] have been utilized, yielding numerous results. However, expensive instruments, time-consuming pre-processing stages, and specialized operations prevent inexpensive, straightforward, and rapid sodium nitrite detection [12]. Meanwhile, it is impossible to disregard the potential contamination risks posed by the employment of detection reagents [13]. As a result, the process of developing new approaches to compensate for the flaws of older systems was a gradual one. Electrochemical approaches have become a promising technique for the detection of sodium nitrite as a result of its low cost, high sensitivity, and simple operation [14]. This is in comparison to the methods that have been described above. Because it does not involve the use of color development reagents, this technique can also be described as being environmentally friendly [13].

Typically, electrochemical methods based on natural enzymes have unrivaled advantages owing to their specificity [15]. However, the primary issue is that the vast majority of naturally occurring enzymes are unreliable when exposed to circumstances such as heat, acids, and bases. They become prone to structural alterations as a result of these severe circumstances, which can ultimately result in the loss of their catalytic activity [16]. Because of this, the research on mimic enzymes came into being, and numerous nanomaterials with peroxidase-like activities, such as carbon-based compounds and metal oxides, have been found and are garnering an increasing amount of interest [17,18]. For instance, Fe_3_O_4_ is a metal oxide that has an activity similar to that of peroxidase. It finds widespread application in electrochemistry, catalysis, and sensor technology [19,20,21]. Due to the high surface free energy and strong magnetic dipole moment interaction of magnetic core Fe_3_O_4_, it is recyclable, and the cost of utilization can be reduced [22]. Nonetheless, they have a tendency to aggregate, which reduces their catalytic activity [23]. Therefore, it is necessary to create magnetic nanocomposite particles by combining magnetic nanoparticles with other substances in order to overcome the limitations of magnetic nanoparticles alone [24,25]. AuNPs have distinct mimetic enzymatic properties compared to other metal nanoparticles, including long-term stability, a large specific surface area, and superior biocompatibility [26]. AuNPs have been used effectively as peroxidase analogues in many electrochemical reactions [27,28,29]. Consequently, the Au magnetic nanocomposites (Au@Fe_3_O_4_), which combine magnetic Fe_3_O_4_ particles and AuNPs, incorporate all the benefits of both materials and can be utilized more broadly.

At the present time, researchers have been applying acoustic chemistry to the synthesis of nanoparticles [30]. To synthesize Au@Fe_3_O_4_, the majority of approaches use sodium citrate or sodium borohydride as a reducing agent, which will result in some environmental contamination [31,32]. It is also known that plant extracts also contain a substantial number of flavonoids, and these flavonoids may be directly engaged in the reduction of precursor metal ions [33,34]. As a result, the application of plant extracts as reducing agents in the process of sonochemical synthesis will either reduce or eliminate the generation of hazardous compounds, which will promote the use of plant extracts as reducing agents in the process of synthesizing nanomaterials. One of the most well-known members of the family Fagaceae, the chestnut is a plant that is commonly farmed in East Asia [35]. During the peeling process of chestnuts, the chestnut skins and interior shells are thrown away. Previous research has demonstrated that chestnut shells contain a significant quantity of flavonoids, which are known for their exceptional antioxidant action [36]. As a result, waste chestnut skin is a fantastic option for a reducing agent that has a great deal of potential for the environmentally friendly synthesis of Au@Fe_3_O_4_.

In this paper, we propose the green synthesis of Au@Fe_3_O_4_ nanoparticles using waste chestnut skin extract to reduce AuNPs and combining it with sonochemical activation. Not only do Fe_3_O_4_ nanoparticles increase the peroxidase-like activity in Au@Fe_3_O_4_, but they also make it possible for Au@Fe_3_O_4_ to be recycled and reused in different configurations depending on the magnetism of the material. In addition, AuNPs that have superior electrochemical characteristics are able to control signal amplification and speed electron transfer, which results in a large increase in the sensitivity and selectivity of electrochemical sensors [37]. As shown in Figure 1, both Fe^3+^ and Au^+^ in a Au@Fe_3_O_4_ peroxidase mimic catalyze H_2_O_2_ to produce oxidatively active free radicals. The free radicals can further catalyze the electronic redox reaction of sodium nitrite on the electrode surface, which enhances the current response in the system and thus can improve the sensitivity of electrochemical detection of sodium nitrite. Based on this principle, an electrochemical detection system for sodium nitrite was built utilizing Au@Fe_3_O_4_ as a peroxidase mimic. This combination of enzyme analog and electrochemical detection methods does not necessitate the modification of electrodes in order to enhance the transmission of electrochemical signals. In the experiment, the optimal electrochemical monitoring system was also determined. The optimal electrochemical sensing system for the experiment has robust anti-interference performance and has been applied effectively to the detection of actual samples. In light of this, the peroxidase-mimic-based electrochemical detection technique will serve as a benchmark in the disciplines of environmental and food analysis.

## 2. Materials and Methods

### 2.1. Materials and Instrumentation

Chestnuts, milk, and kimchi were purchased at Harbin supermarket. Sodium nitrite, potassium chloride (KCl), p-benzoquinone, sodium hydroxide (NaOH), hydrogen peroxide (H_2_O_2_), sodium chloride (NaCl), and phosphate buffer solution (PBS) were all purchased from Tianli Chemical Reagent Co., Ltd. (Tianjin, China). Ferric chloride hexahydrate (FeCl_3_·6H_2_O), ferrous chloride tetrahydrate (FeCl_2_·4H_2_O), glucose, and sucrose were provided by Windship Chemical Reagent Technology Co. (Tianjin, China). Chloroauric acid (HAuCl_4_·3H_2_O), isopropanol, calcium chloride (CaCl_2_), ascorbic acid, ethanol, and lactose were all obtained from Huaian Heyuan Chemical Co., Ltd. (Jiangsu, China). All chemicals used were analytical grade and used according to accepted criteria.

The following instruments were used for the synthesis of Au@Fe_3_O_4_ nanoparticles: KQ3200B ultrasonic cleaning machine (Kunshan Ultrasonic Instrument Co., Jiangsu, China); JD200-3 electronic balance (Shanghai Yiyu Electronic Technology Co., Shanghai, China); PHSJ-3F pH meter (Shanghai Yidian Scientific Instrument Co., Shanghai, China ); SYG-1-2 electric thermostatic water bath (Tianjin Tester Instrument Co., Tianjin, China); L535-1 low-speed centrifuge (Hunan Xiangyi Laboratory Instrument Development Co., Hunan, China). Furthermore, a Model H-7500 transmission electron microscope (Hitachi High-Technologies, Ltd., Tokyo, Japan) was used to reveal the appearance characteristics of Au@Fe_3_O_4_. All electrochemical experiments were performed using a CHI660E electrochemical workstation (Shanghai Chenhua Instrument Co., Shanghai China). A conventional three-electrode electrochemical system was used. Among them, the glassy carbon electrode (GCE) was used as the working electrode, the Ag/AgCl electrode was chosen as the reference electrode, and the platinum electrode was taken as the counter electrode.

### 2.2. Synthesis of Au@Fe_3_O_4_ Nanoparticles

Au@Fe_3_O_4_ was prepared with the sonochemical-activation-assisted biosynthesis method reported by Álvaro de Jesús Ruíz-Baltazar, with some modifications [38]. In detail, at first, the chestnut skin extract was derived from 80 mL of deionized water and 5 g of pre-crushed, soaked chestnut skin. The resulting mélange was heated at 100 °C for 10 min. The chestnut skin extract and the HAuCl_4_·3H_2_O solution were mixed and ultrasonicated at 40 kHz for 15 min. Then, a solution of salt precursors of FeCl_3_·6H_2_O and FeCl_2_·4H_2_O was added to the reaction in a ratio of Fe(Ⅲ)/Fe(Ⅱ) = 2 under ultrasonic agitation for 60 min. In all phases of synthesis, the ultrasonic fluid was operated at a frequency of 40 kHz and 150 W. Finally, the pH of the black liquid obtained was adjusted to 11 with NaOH solution. The Au@Fe_3_O_4_ nanoparticles were washed with isopropanol and desiccated for characterization. 

### 2.3. Verification of Au@Fe_3_O_4_ Peroxidase-Like Activity

In this work, the CV and *i-t* methods were used to evaluate the electrochemical response of Au@Fe_3_O_4_ to sodium nitrite, with reference to and further optimization of the methods of the literature [39]. In the CV method, it is necessary to prepare five different reaction systems to verify the peroxidase-like activity of Au@Fe_3_O_4_ in the electrochemical system: (a) H_2_O_2_ + sodium nitrite, (b) sodium nitrite only, (c) H_2_O_2_ + Au@Fe_3_O_4_, (d) sodium nitrite + Au@Fe_3_O_4_, and (e) H_2_O_2_ + sodium nitrite + Au@Fe_3_O_4_. PBS at pH 6 was used for all reactions. Specifically, we added 0.01 g of Au@Fe_3_O_4_ nanopowder to a small beaker containing 10 mL of PBS (pH 6) and 200 μL of H_2_O_2_ standard solution (50 mmol/L). The reaction system was then heated to a constant 20 °C for 15 min and 200 μL of sodium nitrite standard solution (10 mmol/L) was added for detection.

The *i-t* method required the preparation of two reaction systems to demonstrate the peroxidase-like activity of Au@Fe_3_O_4_ in the electrochemical system: (a) H_2_O_2_ + sodium nitrite and (b) H_2_O_2_ + sodium nitrite + Au@Fe_3_O_4_. The same procedure as for the CV method was used to prepare the reaction system. However, during detection, 200 μL of sodium nitrite standard solution (10 mmol/L) was added to the system every 10 s. The current variation of the oxidation peak current from (a) to (e) was detected using the CV method, and the current variation of (a) and (b) was detected with the *i-t* method.

### 2.4. Electrochemical Detection of Sodium Nitrite

Various masses of Au@Fe_3_O_4_ powder (0.0045, 0.007, 0.0095, 0.012, and 0.0145 g) were added to beakers containing 10 mL of PBS of different pH (4, 5, 6, 7, and 8) and then added to 200 μL of H_2_O_2_ standard solution (50 mmol/L). After all of these were mixed well, the reaction system was heated to different constant temperatures (20, 30, 40, 50, and 60 °C) using a water bath. The water bath was gently shaken every minute during heating. After 15 min of complete reaction, 200 μL sodium nitrite standard solution (10 mmol/L) was added. The electrochemical workstation was adjusted to different scan rates (20, 40, 60, 80, and 100 mV/s) for detection [40]. The CV technique was utilized to detect sodium nitrite in the system with a potential range of −0.8 to 0.8 V. The absolute value of the peak current (|*Ipa*|) generated by the redox reaction of sodium nitrite on the GCE surface was recorded, and the optimal detection strategy was determined.

In the following step, the electrochemical response of sodium nitrite detection was investigated in accordance with the optimal electrochemical detection system derived from optimization experiments. In the concentration range of 0.01–100 mmol/L sodium nitrite, a working curve was constructed with sodium nitrite concentration as the horizontal axis and the absolute value of the oxidation peak current as the vertical axis. On the basis of the obtained working curve, the LOD was determined. Furthermore, we investigated the anti-interference performance of the optimal electrochemical detection system for a number of prevalent substances in food. The selected interfering substances were KCl, NaCl, CaCl_2_, glucose, sucrose, and lactose at a concentration of 1 mol/L. Sodium nitrite at a concentration of 10 mmol/L was used as a reference standard to compare the absolute values of the detected oxidation peak currents to assess the anti-interference of this mimic enzyme detection system. In particular, the concentration of each interfering substance was 100 times higher than the standard solution of sodium nitrite.

### 2.5. Real Sample Analysis

To verify the feasibility of this electrochemical method in real samples, milk, tap water, and kimchi were selected as real samples [41]. Prior to testing, milk and tap water were dissolved in a buffer solution (0.1 mol/L PBS, pH 6) in a 1:10 ratio. Then, 5 g of the kimchi was taken and added to 70 mL of distilled water and extracted with ultrasound for 30 min. The solution was transferred to a 100 mL volumetric flask after 5 min in a 75 °C water bath, diluted to scale with water, and vigorously agitated. After using filter paper to filter the solution, a portion of the effluent was collected in a centrifuge tube. The liquid was centrifuged for 20 min at 4000 rpm. Finally, a certain concentration of nitrite was injected into the actual treated sample, and the sensor was used to detect the nitrite. The absolute values of the oxidation peak currents of three sample solutions with varying concentrations were monitored in accordance with the optimal electrochemical detection system for sodium nitrite. The experiment was repeated to ascertain the accuracy of the sodium nitrite detection system and the recovery of the standard edition.

## 3. Results and Discussion

### 3.1. Characterization of Au@Fe_3_O_4_

The different resolutions of the TEM images of Au@Fe_3_O_4_ are shown in Figure 2a,b. The TEM shows that Au@Fe_3_O_4_ was made up of distributed spheres, with nanoparticles ranging in size from 8 to 15 nm on average. In the meantime, it is clear from Figure 2a that some of the particles gathered together and formed large agglomerates with rough surfaces. The diameter of the large agglomerates was about 200 nm. It is quite likely that the agglomeration phenomena arise as a result of the small size of magnetic nanoparticles combined with their huge specific surface area. The dipole distance between the particles makes it easier for them to combine, which in turn helps to minimize the specific surface energy.

The elemental composition of Au@Fe_3_O_4_ was further analyzed using EDS. In Figure 2c, it can be clearly seen that the assay contains both Fe and Au, and no other elements have been detected. The high-intensity Fe signal indicates the presence of Fe_3_O_4_, while the Au signal is due to the contribution of AuNPs. This is a reliable confirmation that the primary components of Au@Fe_3_O_4_ are AuNPs with magnetic Fe_3_O_4_ and have no other impurities. As a result, using a sonochemical-activation-assisted method, magnetic Fe_3_O_4_ may be effectively combined with AuNPs reduced from the chestnut skin green waste to generate a magnetic nanocomposite, Au@Fe_3_O_4_.

Next, the green-synthesized Au@Fe_3_O_4_ peroxidase mimic was analyzed via infrared spectroscopic analysis, as shown in Figure 2d. According to the results of the FT-IR spectrum, it is clear that the characteristic peak located near 563 cm^−1^ is due to the stretching vibration of the Fe-O bond. The characteristic peak at 1330~1520 cm^−1^ is attributed to C-O stretching. Meanwhile, multiple characteristic peaks at wavelengths between 2150 and 2350 cm^−1^ may be caused by vibrations of long-chain alkyl groups -CH_2_ and -CH_3_. The -OH bond is represented by a characteristic peak at 3828 cm^−1^, which may be formed by the O-H stretching vibration in the -OH with particular catalytic oxidation.

BET characterization was used to further understand the specific surface area and pore data of Au@Fe_3_O_4_. As shown in Table 1, the pore volume of Au@Fe_3_O_4_ was 0.19649 cm^3^/g, and the specific surface area was 60.7859 m^2^/g. In addition, the average pore size of Au@Fe_3_O_4_ was calculated using the Barrett–Joyner–Halenda (BJH) method to be 11.4508 nm, indicating that the Au@Fe_3_O_4_ material has mainly a mesoporous structure. The Au@Fe_3_O_4_ nanoparticles have a large specific surface area and a tiny particle size, allowing them to enhance metal-catalytic active site exposure [42]. By dissolving H_2_O_2_, Fe^2+^ in Fe_3_O_4_ nanoenzyme can produce the active site of ·OH. This means that Fe^2+^ reacts rapidly with H_2_O_2_ to form ·OH and Fe^3+^. Fe^3+^ is slowly reduced to Fe^2+^ by the reaction of the system with the hydrogen donor [43]. Furthermore, gold nanoparticles have been shown to function as peroxidase-like enzymes [44]. The synergistic effect of Au and Fe_3_O_4_ makes the peroxide-mimetic enzymatic activity of Au@Fe_3_O_4_ highly promising.

### 3.2. Peroxidase-Like Activity of Au@Fe_3_O_4_ Nanoparticles

To verify the mimetic enzyme activity of Au@Fe_3_O_4_, five systems were tested using the CV method between −0.8 V and +1.0 V in a 0.1 mol/L, pH = 7 PBS, as shown in Figure 3a. As demonstrated in Figure 3a, the CV method initially provided a weak electrochemical signal when only sodium nitrite was present in the PBS. This can be attributed to the unique electrochemical activity of the sodium nitrite. In PBS, both Au@Fe_3_O_4_ + sodium nitrite and Au@Fe_3_O_4_ + H_2_O_2_ systems displayed oxidation current maxima at 0.52 V, and the current response grew when compared to the presence of sodium nitrite alone. This indicates that Au@Fe_3_O_4_ can catalyze the redox reactions of sodium nitrite and H_2_O_2_ to varying degrees. It has been demonstrated that Au@Fe_3_O_4_ can introduce a significant number of active oxide substances to catalyze the H_2_O_2_ reaction and generate a significant electrical signal [45]. The ability of Au@Fe_3_O_4_ to boost the oxidation process of sodium nitrite in solutions driven by electron transfer shows the peroxidase mimic’s catalytic activity. The H_2_O_2_ + sodium nitrite system showed apparent oxidation current peaks, which may be due to the fact that H_2_O_2_ creates tiny quantities of oxidatively active free radicals in solution, which react with sodium nitrite to generate an oxidation current peak. When H_2_O_2_ + sodium nitrite + Au@Fe_3_O_4_ were present in the system at the same time, the oxidation current peak was visible at 0.52 V, and the current response was substantially greater than in the other systems. Au@Fe_3_O_4_, as a peroxidase mimic, catalyzed the decomposition of H_2_O_2_ and introduced a large number of oxidatively active substances. Therefore, the significantly increased oxidation current peak indicates that the oxidatively active free radicals decomposed by H_2_O_2_ contribute to the oxidation of sodium nitrite to sodium nitrate, which is accompanied by a large number of electron transfers that increase the current response. Au@Fe_3_O_4_ exhibits peroxidase-like enzyme activity that effectively catalyzes H_2_O_2_ and further oxidizes sodium nitrite, and thus it should be used in the electrochemical detection of sodium nitrite. The mechanism of electrochemical oxidation of sodium nitrite is shown in Equations (1)–(5).
(1)$${Fe}^{3+}+e^{-}\to {Fe}^{2+}$$
(2)$${Fe}^{2+}+H_{2}O_{2}\to {Fe}^{3+}+\cdot OH+{OH}^{-}$$
(3)$${Au}^{0}+H_{2}O_{2}\to {Au}^{+}+\cdot OH+{OH}^{-}$$
(4)$${Au}^{0}+H_{2}O_{2}\to {Au}^{+}+\cdot O_{2}^{-}+2H^{+}+2e^{-}$$
(5)$$NO_{2}^{-}+H_{2}O\overset{\cdot OH}{\to }2H^{+}+NO_{3}^{-}+e^{-}$$

The peroxidase-like activity of Au@Fe_3_O_4_ was confirmed further using chronoamperometry, as shown in Figure 3b. When Au@Fe_3_O_4_ nanoparticles and H_2_O_2_ were present in the solution, the current response increased rapidly with the addition of sodium nitrite and reached the steady state current within 2 s. This is because Au@Fe_3_O_4_ catalyzes the generation of a large number of oxidatively active free radicals from H_2_O_2_, and NO_2_^−^ is further catalyzed and loses electrons to oxidize to generate NO_3_^−^, causing a current change. This is because Au@Fe_3_O_4_ catalyzes the production of a substantial number of oxidatively active free radicals from H_2_O_2_, and NO_2_^−^ is further catalyzed, leading to the loss of electrons and the generation of an electric current. The unincorporated Au@Fe_3_O_4_ nanoparticles increased the current less than the incorporated Au@Fe_3_O_4_ system due to the fact that only a tiny number of free radicals are produced when only H_2_O_2_ and sodium nitrite are present, thus confirming further that Au@Fe_3_O_4_ has peroxidase-like activity. The results of both methods demonstrate that Au@Fe_3_O_4_ has peroxidase-like activity and could be effectively used for electrochemically enhanced detection of sodium nitrite to improve detection sensitivity.

### 3.3. Mechanistic Validation of Electrochemical Detection System

As shown in Figure 4, we conducted an investigation into the creation of the absolute value of the oxidation peak currents in the system H_2_O_2_ + sodium nitrite + Au@Fe_3_O_4_ in order to gain an understanding of the fundamentals underlying catalytic oxidation in the system. The breakage of the O-O bond in H_2_O_2_ was facilitated by the presence of Au@Fe_3_O_4_. This resulted in the production of reactive oxygen species (ROS) that possessed oxidative activity. The redox reaction of the sodium nitrite can be seen a large rise as a result of the involvement of ROS. Hydroxyl radicals (·OH) and superoxide radicals (·O_2_^−^), among other types of ROS, were relatively common. Therefore, the oxidation pathway can be validated through the use of tests involving the suppression of free radicals. Ascorbic acid is an effective ROS scavenger that can neutralize a wide variety of free radicals. To begin with, ascorbic acid was introduced to the system so that the presence of ROS could be confirmed. As a result, the current of the oxidation peak in the system was considerably suppressed, and almost no oxidation peak was created at all. This was demonstrated in Figure 4, which can be seen here. This occurrence is proof of the presence of ROS in the process, as well as the formation of a large number of ROS. After that, isopropyl alcohol was utilized as an ·OH scavenger in order to establish that ·OH was being produced. When isopropanol was introduced to the system, there was a discernible drop in the peak value of the oxidation current that was being produced by the system. This is an important finding. As a result, the H_2_O_2_ was converted into a significant quantity of hydroxyl radicals due to the peroxidase mimicry provided by the Au@Fe_3_O_4_ compound. In the end, p-benzoquinone was used as a scavenger of entering ·O_2_^−^ in order to confirm the generation of ·O_2_^−^ that was being introduced. The oxidation peak current of the system was likewise lowered after the addition of p-benzoquinone to the system; however, the magnitude of this reduction was not as great as the reduction that occurred in the system after the addition of isopropanol. Thus, the Au@Fe_3_O_4_ peroxidase mimic catalyzing H_2_O_2_ also produced ·O_2_^−^, and ·O_2_^−^, which can catalyze the oxidation of sodium nitrite. It has also been shown that ·O_2_^−^ is not the main oxidizing agent. Instead, ·OH is the major oxidizing agent of sodium nitrite. To sum up the above arguments, Au@Fe_3_O_4_ nanoparticles as a peroxidase mimic promote a variety of ROS generation and electron transfer. It is mainly ·OH that promotes the oxidation reaction of sodium nitrite. For the electrochemical detection system of sodium nitrite, Au@Fe_3_O_4_ was shown to have good peroxidase-like activity.

### 3.4. Optimization of Electrochemical Detection Conditions for Peroxidase Mimic

The effects of Au@Fe_3_O_4_ addition, reaction temperature, pH, and scan rate on the electrochemical response were explored and optimized in order to provide a greater level of sensitivity and performance in the electrochemically enhanced detection of sodium nitrite. This was accomplished by determining the optimal values for each of these variables. In order to determine the detection index, the absolute value of the oxidation peak current was utilized.

The electrochemical detection system is subject to the direct influence of the temperature of the reaction. As can be observed in Figure 5a, the absolute value of the oxidation peak current progressively increases as the reaction temperature rises. This trend is depicted by a linear trend line. When the temperature was in the range of 20 to 30 °C, there was a quick increase in the response rate, as well as an increase in the number of charges on the electrode surface. As a direct consequence of the quickening of the electron transfer rate and the intensification of the current response signal, the absolute magnitude of the oxidation peak current rose precipitously in a short period of time. The temperature range of 30–50 °C was optimal for maximizing the peroxidase-like activity of the Au@Fe_3_O_4_ enzyme in the assay method. The ongoing oxidative breakdown of H_2_O_2_ resulted in the production of a significant amount of ROS, which in turn sped up the electrocatalytic oxidation of sodium nitrite. The electrochemical response signal became stronger as the number of electron transfers on the electrode surface rose, and the absolute value of the oxidation peak current increased as the number of electron transfers on the electrode surface increased. However, when the temperature was increased even further to the range of 50–60 °C, the reaction rate dropped dramatically, the electrochemical signal became less strong, and the absolute magnitude of the oxidation peak current showed a downward trend. It was determined that this was due to the fact that when the temperature was higher than 50 °C, the Au@Fe_3_O_4_ peroxidase mimic experienced some inhibition and a drop in its catalytic activity. This led to a large reduction in the amount of ·OH that was created as a result of the catalytic reaction. The temperature at which the electrochemical sensor began to react was also too high to be easily recognized. As a result, the temperature of 50 °C was determined to be the most suitable for the system’s reactions.

The scan rate was crucial to the system’s electrochemical response. As shown in Figure 5b, the absolute value of the system’s oxidation peak current increased dramatically as the scan rate was gradually increased. Excellent linearity existed between the absolute value of the oxidation peak current and the square of the root of the scan rate. The regression equation was *y* = 87.35*x* + 3.877 (*R*^2^ = 0.9907). This was because the diffusion effect controls the electron transfer process in the system. As the scan rate increases, so does the electron transfer rate, and, as a result, the absolute value of the oxidation peak current increases [46]. Nevertheless, our group’s previous experimental investigations demonstrated that when the scan rate exceeded 0.1 V/s, the stability of the detection system was compromised, which was not conducive to further sodium nitrite detection. As a result, 0.1 V/s was chosen as the optimal scan rate.

In addition, the detection system’s Au@Fe_3_O_4_ addition was optimized. As depicted in Figure 5c, the additional quantity of Au@Fe_3_O_4_ was between 0.0045 and 0.0095 g, and the absolute value of the oxidation peak current increased gradually as the additional amount was added. With the addition of more Au@Fe_3_O_4_ to the electrochemical detection system, the peroxidase-like activity likely increased, promoting the electrooxidation of sodium nitrite to generate electron transfer and generate more ·OH. It was also possible that a trace quantity of Au@Fe_3_O_4_ was attached to the surface of the electrode, which boosted the sensitivity of the electrode and the electron transfer rate significantly. Thus, when the amount added was between 0.0095 g and 0.0145 g, the reaction rate decreased swiftly, and the absolute value of the oxidation peak current decreased gradually. It was conceivable that the nanoparticle aggregation caused by the excessive addition of Au@Fe_3_O_4_ reduced the catalytic effect. The optimal amount of Au@Fe_3_O_4_ to be added for the detection of sodium nitrite was determined to be 0.0095 g.

The results of the effect of buffer pH on the electrochemical detection system are shown in Figure 5d. With the increase in buffer pH, the absolute value of the oxidation peak current first increased and then decreased. At pH 6 of the test system, the peak oxidation current intensity was maximum. This is because the electrons were easily transferred to the electrode surface by the oxidation of sodium nitrite in the pH 6 buffer. A stronger electrochemical signal was generated, and the absolute value of the oxidation peak current increased accordingly. In the meantime, the medium has a significant quantity of negative ions when the pH of the buffer solution is in the range of 7 to 8. The presence of an excessive number of negative ions fails to promote the flow of electrons to the electrode, and the electron mobility gradually diminishes as a result. This makes it challenging for the system to go through redox reactions. The electrochemical reaction will be slowed down, and the absolute value of the peak current produced by oxidation will decrease. As a consequence, pH 6 was chosen as the electrochemical condition for further measurement of sodium nitrite [47].

Overall, the optimal combination of experimental schemes was a reaction temperature of 50 °C, an amount of added Au@Fe_3_O_4_ of 0.0095 g, a scanning rate of 0.10 V/s, and a buffer pH of 6. Three parallel tests were performed under optimal conditions, and the average absolute value of the oxidation peak current was approximately 14.27 μA.

### 3.5. Electrochemical Detection of Sodium Nitrite

As illustrated in Figure 6a, it can be observed that the current response of the oxidation peak increases proportionally with the rise in sodium nitrite concentration. It can be seen from Figure 6b that the concentration of sodium nitrite has a good linear relationship with the absolute value of the oxidation peak current in the range of 0.01–100 mmol/L. The regression equation of the working curve was *y* = 1.0752*x* + 4.4728 (*R*^2^ = 0.9949). The calculated LOD was 0.867 µmol/L (*S/N* = 3). The numerous electrochemical sensors that can detect sodium nitrite are compared and contrasted in Table 2, which can be found below. This system had a lower LOD than many of the previously described methods, a wider linear range than other nanomaterial-based electrochemical assays, and good sensitivity. Additionally, this system had a lower linear range.

To evaluate the anti-interference performance of the system, six prevalent food and environmental contaminants were selected as interference substances. As depicted in Figure 7a, the electrochemical detection system exhibited a negligible current response to interfering substances. In contrast, it exhibited a notable current response to sodium nitrite with a high absolute value for the oxidation peak current because the presence of ·OH in the system catalyzes the reaction of NO_2_^−^ to generate NO_3_^−^, producing a variation in the current and an increase in the absolute value of the oxidation peak. While other interference substances are not able to respond, there is almost no current change. As a consequence, the electrochemical detection system for sodium nitrite has excellent anti-interference performance. The stability of the sensor was investigated by performing 30 parallel measurements on sodium nitrite (10 mmol/L). The sensor does not use modified electrodes and can be reused for testing after rinsing. The sensor was detected in a buffer solution of 0.1 mol/L PBS (pH 7) over a range of −0.8 to +0.8 V with CV. As shown in Figure 7b, the stability of the sensor remained stable over 96.59%, which indicated the excellent stability of the electrode.

### 3.6. Real Sample Analysis

The electrochemical method’s viability in real samples was evaluated, including milk, tap water, and kimchi, to determine how well it performed. The 10 mmol/L sodium nitrite standard solution was chosen and evaluated using the most effective detection technique. Table 3 displays the results of a recovery evaluation based on the regression equation of the produced working curve. RSDs were all below 4%, with calculated recoveries ranging from 97.79% to 100.54% for milk, 98.64% to 104.25% for tap water, and 99.31% to 102.47% for kimchi. Thus, the sodium nitrite system based on the Au@Fe_3_O_4_ peroxidase mimic offers good accuracy and sensitivity in electrochemical detection. The superior repeatability of the upgraded electrochemical detection equipment is evident. In summary, the established electrochemical detection based on the peroxidase-like activity of Au@Fe_3_O_4_ has great potential for sodium nitrite detection.

## 4. Conclusions

In summary, based on the Au@Fe_3_O_4_ peroxidase-like activity, a novel electrochemical sensor for the efficient real-time detection of sodium nitrite has been successfully designed and fabricated in this work. The reduction of chloroauric acid by chestnut skin was supported by sonochemical activation and combined with Fe_3_O_4_ magnetic nanoparticles to generate Au@Fe_3_O_4_. The Au@Fe_3_O_4_ can catalyze H_2_O_2_ to produce ROS for the redox reaction of sodium nitrite. On this basis, the electrochemical detection of sodium nitrite was achieved. Compared to other detection methods with modified electrodes, it is more suitable for real-time detection. And the addition of the Au@Fe_3_O_4_ peroxidase mimic increases the oxidation peak current and reduces the error, making the sensor stable and maintaining the stability over 96.59%. In addition, the electrochemical sensor is highly resistant to coexisting molecules. There were also satisfactory recoveries in the actual sample analysis, demonstrating the usefulness of the sensor. Admittedly, the established sodium nitrite assay, based on the activity of the Au@Fe_3_O_4_ peroxidase mimic, makes the rapid detection of sodium nitrite more versatile and has potential applications.

## Figures and Tables

**Figure 1 foods-12-03665-f001:**
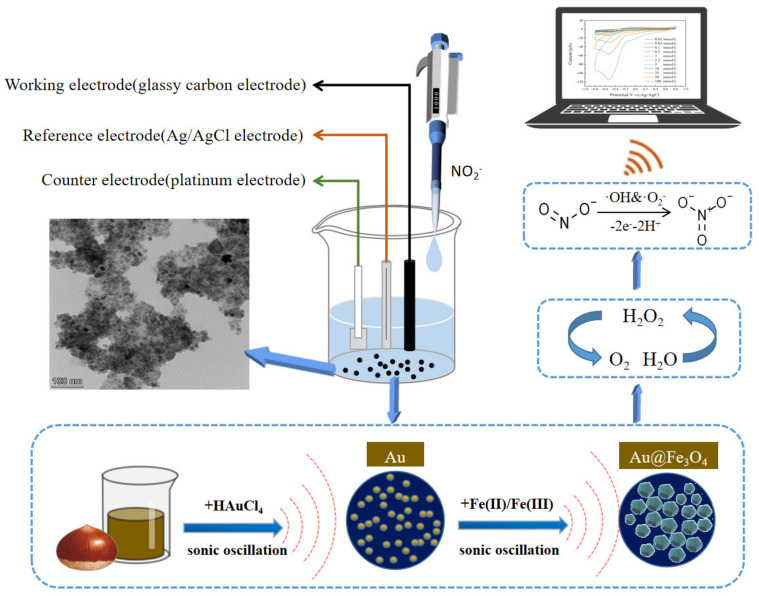
Schematic diagram of the sonochemical green synthesis of Au@Fe_3_O_4_ and its electrochemical detection of sodium nitrite as a peroxidase mimic.

**Figure 2 foods-12-03665-f002:**
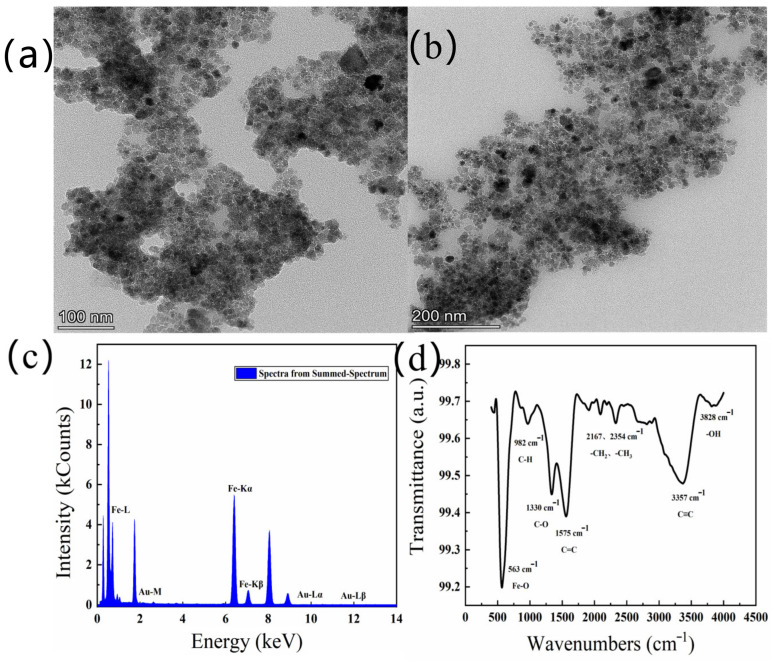
TEM images of Au@Fe_3_O_4_ with the scale bar 100 nm (**a**) and 200 nm (**b**), EDS diagram of Au@Fe_3_O_4_ (**c**), Fourier infrared spectrum of Au@Fe_3_O_4_ (**d**).

**Figure 3 foods-12-03665-f003:**
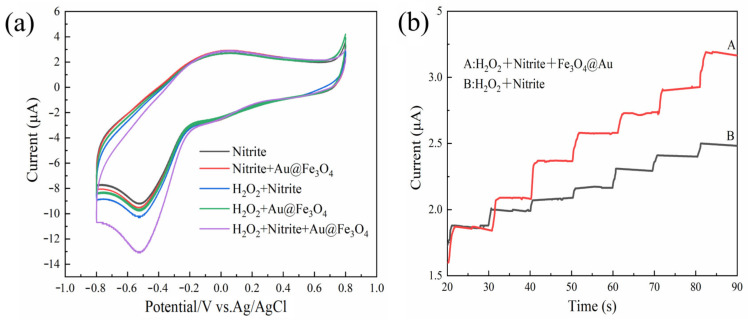
(**a**) Verification of peroxidase-like activity of Au@Fe_3_O_4_ by the CV method; (**b**) verification of peroxidase-like activity of Au@Fe_3_O_4_ by the *i-t* method, A: the system containing peroxidase mimic, and B: the system without peroxidase mimic.

**Figure 4 foods-12-03665-f004:**
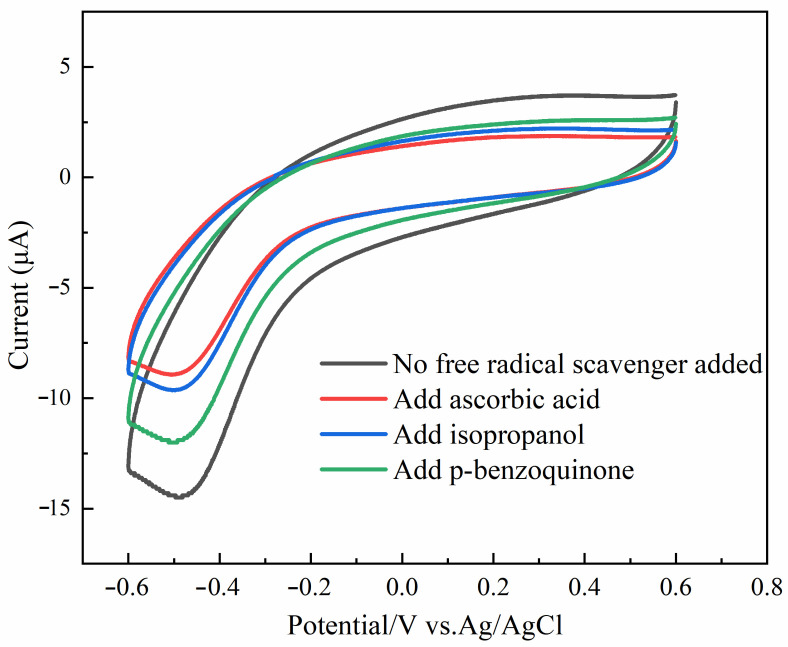
Mechanistic validation of Au@Fe_3_O_4_ peroxidase mimic detection of sodium nitrite.

**Figure 5 foods-12-03665-f005:**
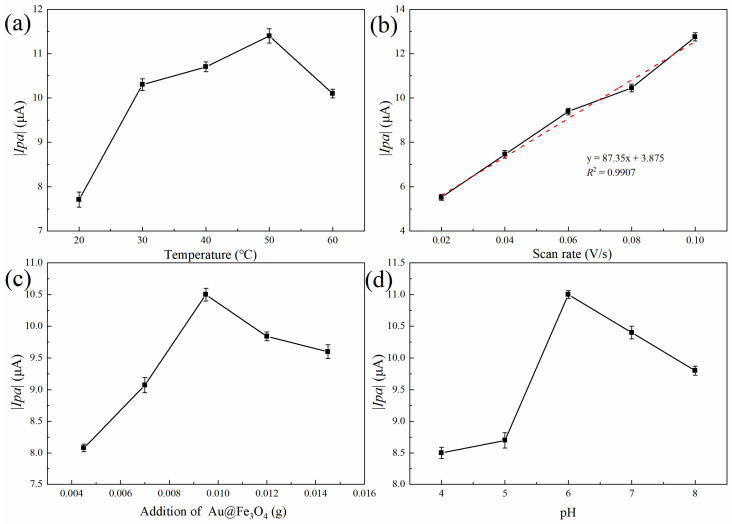
Optimization of Au@Fe_3_O_4_ peroxidase mimic electrochemical detection conditions: (**a**) effect of reaction temperature on the electrochemical detection system, (**b**) peak current vs. square root of scanning rate, (**c**) effect of Au@Fe_3_O_4_ addition on the electrochemical detection system, and (**d**) effect of pH on the electrochemical detection system.

**Figure 6 foods-12-03665-f006:**
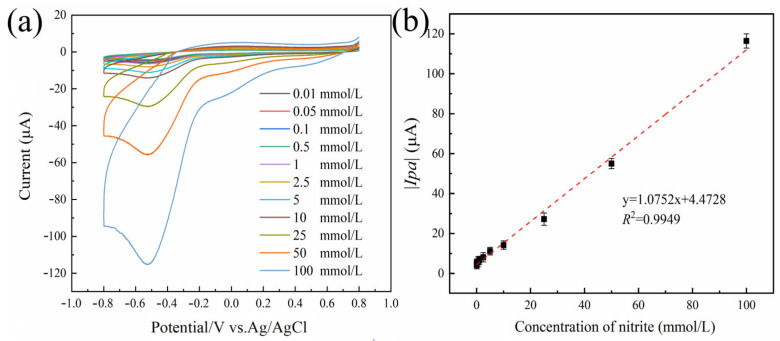
The response curve of Au@Fe_3_O_4_ peroxidase mimic system for the detection of sodium nitrite (**a**). Linearity of the absolute value of the oxidation peak current versus sodium nitrite concentration in the range of 0.01 to 100 mmol L^−1^ (**b**). Error bars indicate the standard deviation (SD) of the repeated measurements at *n* = 3.

**Figure 7 foods-12-03665-f007:**
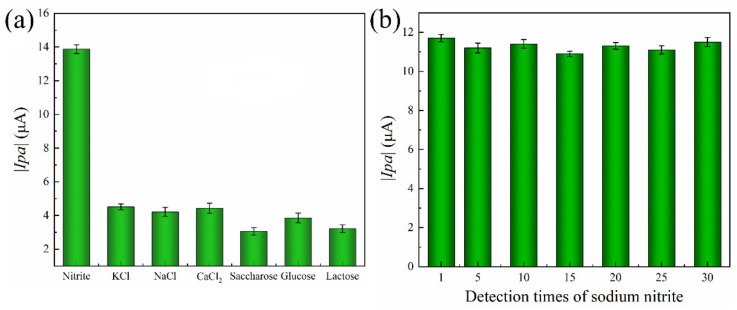
(**a**) Anti-interference and (**b**) stability of Au@Fe_3_O_4_ peroxidase mimics for electrochemical detection nitrite detection system.

**Table 1 foods-12-03665-t001:** BET analysis of Au@Fe_3_O_4_.

	Au@Fe_3_O_4_
pore volume (cm^3^/g)	0.19649
specific surface area (m^2^/g)	60.7859
average pore size (nm)	11.4508

**Table 2 foods-12-03665-t002:** Comparison of different electrochemical sensors for the detection of sodium nitrite.

Electrochemical Sensors	Method	Linear Range (μmol/L)	LOD (μmol/L)	Reference
ɑ-MnO_2_-MoS_2_	CA	100–800	16	[48]
Pt-Cu/GO	*i-t*	3–9000	3	[49]
Ag-Cu@ZnO	LSV	0–1500	17	[50]
TiO_2_-Ti_3_C_2_TX/CTAB/CS/GCE	DPV	3–1250	0.85	[51]
AuNPs/GCE	DPV	1–3800	2.4	[52]
AgNC@NCS	DPV	1.12–1400	0.38	[53]
PrFeO_3_-MoS_2_	CV	5–300	1.67	[54]
MnO_2_/PANI/GCE	CV	100–10,000	4.38	[55]
MWCNTs/PPy-C/GCE	*i-t*	100–800	5–9500	[56]
PEDOT-HMF	LSV	50–7500	0.59	[57]
PAR/Fe_3_O_4_/GCE	CV	9.64–1300	1.19	[58]
CS/MWCNTs/CNs/GCE	LSV	5–1000	0.89	[59]
Au@Fe_3_O_4_	CV	0.01–100	0.867	This work

**Table 3 foods-12-03665-t003:** Actual sample testing.

Sample	Added (mmol/L)	Found (mmol/L)	Recovery (%)	R.S.D (%)
Milk	0	0.000	0	-
	2.5	2.49 ± 0.0276	99.68 ± 0.0143	2.89
	25	24.44 ± 0.0128	97.79 ± 0.0085	3.37
	50	50.27 ± 0.0223	100.54 ± 0.0097	2.22
Tap water	0	0.004	0	-
	2.5	2.466 ± 0.0151	98.64 ± 0.0054	3.41
	25	26.063 ± 0.0278	104.25 ± 0.0112	1.96
	50	50.665 ± 0.0366	101.33 ± 0.0256	1.24
Kimchi	0	0.003	0	-
	2.5	2.51 ± 0.0132	100.38 ± 0.0079	2.93
	25	25.618 ± 0.0087	102.47 ± 0.0035	2.29
	50	49.655 ± 0.0229	99.31 ± 0.0067	1.97

## Data Availability

The data that support the findings of this study are available from the corresponding author upon reasonable request.

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
