# Peer review of "Green Synthesis of Au Magnetic Nanocomposites Using Waste Chestnut Skins and Their Application as a Peroxidase Mimic Nanozyme Electrochemical Sensing Platform for Sodium Nitrite"

_foods, 2023, doi:10.3390/foods12193665_

Round 1

Reviewer 1 Report

The manuscript foods-2490714 entitled "Green synthesis Au magnetic nanocomposites using the waste chestnut skins and their application as peroxidase mimic nanozyme electrochemical sensing platform for sodium nitrite" reports synthesizing an electrochemical sensing platform made of waste chestnut skins stabilized Au magnetic nanocomposites for the determination of sodium nitrite. Application of electrochemical sensors for food monitoring is important way in analytical science. Modified sensors are good choices in this major and this paper focused on it. Manuscript contains some interesting observations, however, there are several additional comments regarding this manuscript:

Abstract must be more discuss and more improve

Keywords→ arrange in alphabetical order.

Procedures in experimental section must be suitable reference if possible

Materials and Methods: In this paper, Materials and Methods need to be improved. Add information about instruments (Model, country)

Tafel investigation should be add.

Stability of sensor must be added in revise version

Please improve the literature survey with more recent works such as:

10.1134/S102319351610013X

In this paper, the methodology is well written, but discussion need to be improved. Results presented need a better discussion. There was no enough discussion or analysis of the results. There was no enough discussion or analysis of the results. The author should explain and clearly discuss this part based on scientific knowledge. It is better to compare the results with more similar recent works. And discuss the superiority of the work.

Reviewer 2 Report

Green synthesis Au magnetic nanocomposites using the waste chestnut skins and their application as peroxidase mimic nanozyme electrochemical sensing platform for sodium nitrite

Comment #01: Elementary mapping (SEM) is needed to ensure that AuNP and Fe3O4 NP are distributed homogeneously.

Comment #02: In Figure 3A, point the reference electrode at the X-axis.

Comment #03: In Figure 3a, what is the CV in the background? And what is the supporting electrolyte and its concentration? It needs to be clear.

Comment #04: In Figure 3b, why does 0 mmol/L have a current of 1.8 mA? To what do you attribute it? What concentration varies and what analyte is it?

Comment #05: In Fig 3 it´s no clear that it varies if it is nitrite or hydrogen peroxide?

Comment #06: The description of Fig 3, in line 283 the information is very thin, improve text to make it clear to readers

Comment #07: Fig. 4 clearly demonstrates that your sensor is not selective and not suitable for complex matrices.

Comment #08: The correlation factor is very low, do you think it is analytically correct? Where are the CVs that have this information?

Comment #09: Cyclic voltammetry is not the most selective technique to perform a calibration curve, why did I choose it? (Fig. 6)

Comment #10: constant typo of "Potnial" and should say "Potential", it is constant in all figures

Comment #11: What are the concentrations of nitrite and its interferents, what technique do I use? Is it ideal to add electroanalytical analysis?

Comment #12: Tables 1, 2 and 3 do not correspond to the complementary information, since they are in the text, please remove the tables from the complementary information, if there is any complementary information add it where appropriate.

Comment #13: in general, your manuscript is not easy to understand, please correct it.

Dear editor:

It has many typographical errors.

Reviewer 3 Report

The manuscript focused on using an electrochemical sensing platform for nitrite detection. It is written nicely and presented very well. It can be accepted after major revision and answers to the queries raised:

1. The overall manuscript lacks novelty in terms of nanoparticle synthesis as well as sensor fabrication.

2. The manuscript has various typo errors (e.g Fig 3 a, X axis write potential) and requires extensive grammatical revision.

3. Why oxidation peak current is not increasing in CV graphs?

4. How the developed sensor can be employed in the detection of real samples, as current readings can not be same in every experiments. So how do you finalize the current range in real samples detection?

Minor editing of English language required

Reviewer 4 Report

The presented article is interesting and deserves good consideration. I have the following questions: 

  1. Is it a mistake in the word “potental” that assumed to be potential in X axes of figures (at least fig. 3,4)? 
  2. Could you add discussion (or specify if it is in text) by which mechanism the selectivity and good anti interference stability is provided? Why system is selective for nitrates and other molecules do not interfere? 
  3. How the improvement in stability of system was measured? (This is stated in conclusion) 

The English quality is at acceptable level. 

Round 2

Reviewer 1 Report

No Comment

Author Response

We are truly grateful to yours initial review and constructive comments on the manuscript. 

Reviewer 2 Report

Dear Guan H. et al:

Comment #01: Elementary mapping (SEM) is needed to ensure that AuNP and Fe3O4 NP are distributed homogeneously (not previously answered)

Comment #02: inform that I use 0.01 mmol/L as a supporting electrolyte. However, it reports that the calibration curve is 1*10-5 at 0.1 mmol/L, so the highest concentrations are higher than the PBS concentration, so it will not be able to compensate the load.

comment #03: Figures 2, 3, 4 and 6 in the cyclic voltammetry on the X axis should say "Potential/V vs reference electrode". Reference electrode: Ag/Cl/Ag, SHE, RHE, NHE, among others

Dear editor:

the quality of english is good.

Reviewer 3 Report

The manuscript has been revised well and can be accepted in its present form.

Author Response

(The authors gave the same response as above.)

Reviewer 4 Report

I can't recommend this publication to be accepted. I did not find answers to my questions substantial. Moreover, corrected version contains new mistakes such as: 

sentence (page 2) regarding flavonoids: 'surprisingly' is not a proper word here since it is well-known that there are flavonoids in the plant extracts and there are so many methods of synthesis based on extracts

abstract has been started with word 'hence' 

again in conclusion there are sentences that are not supported by experimental evidence 

also it is not clear how sensor mimics the enzymes 

the keyword 'gold magnetic particles' also does not seem as appropriate 

how detection stability was calculated? 

Quality of english decreased after revision. 
